# Research on Chinese Nested Entity Recognition Based on IDCNNLR and GlobalPointer

**Weijun Li** [1,2,*] **, Jintong Liu** [1] **, Yuxiao Gao** [1] **, Xinyong Zhang** [1] **and Jianlai Gu** [1]

1   School of Computer Science and Engineering, North Minzu University, Yinchuan 750021, China; 15632418501@163.com (J.L.)
2   State Ethnic Affairs Commission Key Laboratory of Graphic Image Intelligent Processing, North Minzu University, Yinchuan 750021, China
*   Correspondence: lwj@nmu.edu.cn

**Abstract:** The task of named entity recognition (NER) is to identify entities in the text and predict their categories. In real-life scenarios, the context of the text is often complex, and there may exist nested entities within an entity. This kind of entity is called a nested entity, and the task of recognizing entities with nested structures is referred to as nested named entity recognition. Most existing NER models can only handle flat entities, and there has been limited research progress in Chinese nested named entity recognition, resulting in relatively few models in this direction. General NER models have limited semantic extraction capabilities and cannot capture deep semantic information between nested entities in the text. To address these issues, this paper proposes a model that uses the GlobalPointer module to identify nested entities in the text and constructs the IDCNNLR semantic extraction module to extract deep semantic information. Furthermore, multiple-head self-attention mechanisms are incorporated into the model at multiple positions to achieve data denoising, enhancing the quality of semantic features. The proposed model considers each possible entity boundary through the GlobalPointer module, and the IDCNNLR semantic extraction module and multi-position attention mechanism are introduced to enhance the model's semantic extraction capability. Experimental results demonstrate that the proposed model achieves *F1* scores of 69.617% and 79.285% on the CMeEE Chinese nested entity recognition dataset and CLUENER2020 Chinese fine-grained entity recognition dataset, respectively. The model exhibits improvement compared to baseline models, and each innovation point shows effective performance enhancement in ablative experiments.

**Keywords:** named entity recognition; natural language processing; deep neural networks; feature extraction; knowledge graph

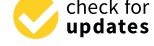

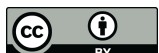

## 1. Introduction

Named entity recognition (NER) is an important task in natural language processing (NLP) [1] and information extraction, aiming to identify named entities in text and predict their types. NER serves as a necessary component for various NLP tasks such as relation extraction, syntax parsing, and knowledge question answering. The task provides entity information for these higher-level tasks. However, named entities exhibit subjectivity, complexity, and variability, and the contextual features of named entities are sparse. As a result, NER has always been a challenging research topic in NLP.

For example, in the sentence "李健平常与王林通电话" ("Li Jianping usually talks on the phone with Wang Lin"), under normal circumstances, "李健平" ("Li Jianping") should be recognized as a person entity, as "李健平" is the subject of the sentence. However, in some special cases, there may exist a person named "李健" ("Li Jian"), and the true meaning of the sentence is that "李健" usually talks on the phone with Wang Lin. In this case, the entity recognition model should identify "李健" as the person entity.

Based on whether entities can overlap in the text, entity recognition can be divided into flat entity recognition and nested entity recognition. In flat entity recognition tasks, entities do not share any overlapping characters. Each character is either an invalid character or belongs to a unique entity. For example, in the text "北京有故宫" ("Beijing has the Forbidden City"), there are only two entities, "北京" ("Beijing") and "故宫" ("Forbidden City"), and these two entities do not share any characters. Therefore, the entity recognition task for this text can be achieved through a flat entity recognition model. However, in reality, text context is complex, and there may exist nested entities within an entity. For example, in the text "北京大学坐落在海淀区" ("Peking University is located in Haidian District, Beijing"), there are three entities, "北京大学" ("Peking University"), "北京" ("Beijing"), and "海淀区" ("Haidian District"). In this case, the entity "北京大学" contains the nested entity "北京". Thus, in complex contexts, nested entities occur more frequently, and the entity recognition task for such texts cannot be achieved through a flat entity recognition model. To address this issue, nested entity recognition methods have been developed, allowing various types of entities to exist within an entity. For example, in the aforementioned text "北京大学坐落在海淀区" ("Peking University is located in Haidian District, Beijing"), the organization entity "北京大学" ("Peking University") contains the location entity "北京" ("Beijing").

With the emergence of the neural network model AlexNet, deep learning gained significant popularity, and various classic neural network modules were proposed. Along with the popularity of deep learning, many researchers began to focus on deep learning-based models for natural language processing. NER systems based on deep learning have achieved remarkable performance improvement. Many studies have used deep learning models to enhance the feature extraction layer of NER models, allowing them to extract more semantic information from text. Dai et al. [2] added an attention mechanism to improve the quality of semantic features. Zhang et al. [3] added deep belief network (DBN) to the entity recognition model, which improved the classification accuracy of the model. Wang et al. [4] used the BILSTM-CRF model to realize entity recognition, and obtained text timing information through BILSTM. Robert et al. [5] added a CNN module to improve the local feature extraction capability of the entity recognition model. Therefore, this paper adopts an entity recognition model based on deep learning to implement the nested entity recognition task. As the feature capacity of classical neural network models is insufficient, this paper constructs a deep semantic extraction module called IDCNNLR. To enhance feature quality, this paper incorporates multi-head self-attention mechanisms at multiple positions in the model. Finally, the model predicts nested entities and their categories through the GlobalPointer module. The paper proposes a Chinese nested entity recognition model called BERT-ATT-IDCNNLR-ATTR-GlobalPointer.

Currently, the medical field contains a large amount of information, covering a wide range of content. It is necessary to perform the timely processing of disease, medication, and medical device information. Qi et al. [6] realized entity recognition of electronic medical records through Multi-Neural Network Fusion. Wang et al. [7] integrated a knowledge graph and attention mechanism into the entity recognition model to realize entity recognition of Internet medical consultation texts. Tan et al. [8] used a pre-trained language model to implement entity recognition in clinical medical texts. In real life, text contexts are complex, and commonly used texts like medical data inevitably contain numerous nested entities. To validate the nested entity recognition capability of our model on medical information, the paper selected the CMeEE dataset as the first experimental dataset. The CMeEE dataset is a nested entity recognition dataset for Chinese medical texts, categorizing medical named entities into nine classes: diseases (dis), clinical symptoms (sym), medications (dru), medical equipment (equ), medical procedures (pro), body parts (bod), medical examination items (ite), microorganisms (mic), and departments (dep). By performing nested entity recognition on medical information, medical institutions can efficiently extract useful entities from medical data, thereby accelerating the operational efficiency of medical institutions.

Currently, the content of real-life texts covers a wide range, and a paragraph of text may contain various types of data. To validate the entity recognition capability of our model on everyday life texts, the paper selected the CLUENER2020 dataset as the second dataset. The Chinese fine-grained named entity recognition dataset CLUENER2020 is based on the THUCTC text classification dataset released by Tsinghua University, with a portion of the data selected for fine-grained named entity annotation. CLUENER2020 includes a total of 10 label types: organization, name, address, company, government, book title, game, movie, organization institution, and scenic spot. The "company" and "government" categories are quite similar and both belong to the "institution" category. These two categories are not as easily distinguishable as the "address" and "institution" categories. Therefore, this dataset can verify the model's ability to differentiate similar fine-grained categories. In entity recognition, the number of entity categories is about ten, and this entity recognition task is called fine-grained entity recognition.

This paper is divided into five chapters: introduction, related work, model, experiments, conclusion, and future work. In terms of structure, the introduction and related work sections first present the current status and significance of nested entity recognition research. Then, the model section describes the specific details of the model. The effectiveness of the model is analyzed through experiments. Finally, the conclusion and future work sections summarize the main contributions of this paper and provide prospects for future research in nested entity recognition.

## 2. Related Work

Entity recognition methods can be categorized into three types: sequence labeling-based methods, hypergraph-based methods, and span-based methods. Sequence labeling-based methods assign labels to each character in the text to predict the entities and their categories.

Common sequence labeling methods include "BIO" and "BMES" methods. For example, in the "BIO" labeling method, "B-Category" represents the beginning character of an entity, "I-Category" represents the internal characters of an entity, and "O" represents non-entity characters. Early sequence labeling methods assigned only one label to each character, making it impossible to assign corresponding labels to nested entities. To address this issue, researchers have proposed multi-label sequence labeling methods that allow a character to have multiple entity labels. These methods recognize nested entities through the multi-label annotations of characters. To enhance the capability of sequence labeling methods in nested entity recognition, researchers have proposed three strategies: hierarchical, cascaded, and structured (joint labeling) labeling approaches [9].

The hierarchical approach overlays a flat entity recognition layer based on the hierarchical information of nested entity structures, and multiple hierarchical entity recognition layers are used to recognize nested entities within entities. Ju et al. [10] dynamically stacked multiple flattened feature extraction layers (BiLSTM+CRF) in the decoding part of entity recognition, and the study uses a hierarchical entity recognition model to recognize nested entities in biomedical texts. However, this stacking approach may introduce error propagation in the model structure, and the data in the model can only be transmitted in one direction, making it unable to utilize the information of outer entities when recognizing inner entities. Wang et al. [11] proposed a pyramid NER model, which is also constructed based on the hierarchical approach. This model first stacks characters towards entity mentions from bottom to top to obtain the forward pyramid structure of entity mentions, which represents all possible entity fragments in the text. Then, the model decodes the forward pyramid in a reverse hierarchical manner to recognize all nested entities in the text.

The cascaded approach builds a binary classifier for each entity category to predict nested entities and their categories. Shibuya et al. [12] viewed the label sequence of nested entities as the second-best path within the long entity scope, and iteratively decoded the path from outer to inner layers to extract inner entities. This model used a single-layer CRF as the classifier for each cascade, and the CRF layer predicted the entity category. This

method combines the hierarchical and cascaded approaches but also suffers from error propagation between layers, which can affect model performance.

The structured strategy predicts each nested entity separately and assigns a separate character label to each nested entity. This method assigns multiple nested entity character labels to characters in the text. For example, in the text segment "北京大学" (Peking University), the character "北" ("north") is the first character of both entities "北京" (Beijing) and "北京大学" ("Peking University"), so the character "北" would be labeled as "B-ORG" and "B-LOC", indicating the first character of an organization entity and a location entity, respectively. Straková et al. [13] used a BERT layer as the word embedding layer to extract word-level information from the text. The study assigned multiple entity labels to each character in the text and introduced other semantic features such as part-of-speech tags to improve feature quality. However, this model requires a significant amount of human resources to annotate transformations in the corpus and perform complex feature engineering when dealing with nested entities. Additionally, expanding the label set in this way may lead to label sparsity issues and reduce the model's recognition ability. This is an introduction to the sequence-based approach.

The hypergraph-based approach constructs a hypergraph of a sentence, with each character as a node and the connections between characters as edges, based on the nested entity structure information. This approach performs well in recognizing nested entities and avoids the complexity issue of the structured strategy's multi-labeling problem. However, in cases where the sentence is too long or there are numerous entity categories, the hypergraph in this approach can become very complex, which can impact model performance. Lu et al. [14] proposed a hypergraph-based model that utilizes the hypergraph to represent semantic information between characters and performs entity boundary detection and category prediction through hypergraph computation. By processing the sub-hypergraph structure of a sentence, this model can recognize nested entities with infinitely long text fragments. However, this model may identify erroneously nested entities, where one entity is contained within another entity, the boundaries of the two entities do not overlap, and both entities have the same type. In such cases, the model may calculate ambiguous structural representations, leading to incorrect predictions. To address this issue, Wang et al. [15] introduced a neural segment hypergraph model that obtains distributed feature representations with neural networks. This model reduces the probability of computing ambiguous structural representations but may have longer training and inference times and higher time complexity compared to the previous model. Katiyar et al. [16] used a recursive neural network to extract features from directed hypergraph representations and constructed an LSTM-based sequence labeling model to learn the hypergraph representation of nested entities in the text. This model performs multi-label predictions on characters during hypergraph construction and uses LSTM module to learn the features of the hypergraph. The model finds the sub-hypergraph with the highest confidence, which represents all nested entities in the text. The hypergraph-based methods can effectively represent and recognize nested entities, reducing the complexity of models that assign multiple labels to a character.

The span-based methods enumerate all possible entity boundaries and then perform recognition and classification on the entity boundaries. Yi et al. [17] concatenated feature vectors of entity boundaries to obtain entity boundary representations using pre-trained models. These representations were then fed into fully connected layers or a biaffine mechanism for entity classification, thus identifying all nested entities and their types. Building upon the previous research, Sohrab et al. [18] set a maximum length for entity boundaries and enumerated all possible entity boundaries. The study used a bidirectional LSTM module to extract semantic features from the entity boundaries and predicted and classified the entity boundaries based on their feature vectors, thus recognizing all nested entities and their types in the text. Chen et al. [19] proposed the NNBA model, which separately extracts the starting and ending boundaries of entities. The model combines the starting and ending boundaries of entities using a forward matching algorithm and predicts and

classifies all possible entity boundaries using a final classifier. Lin et al. [20] used a head-driven phrase structure for nested named entity recognition. Xia et al. [21] proposed an MGNER neural network model that predicts each possible entity boundary, extracts all nested entities, and then classifies the recognized entities using a classification network module. While these methods can recognize nested entities, they have drawbacks such as high computational cost, insensitivity to boundary information, underutilization of partially matching entity boundary information, and difficulty in identifying entities in long texts. To address these issues, Shen et al. [22] divided the entity recognition task into two stages: entity boundary filtering and entity category prediction. In the first stage, this method extracts the boundaries of nested entities through filtering and boundary regression of seed spans. In the second stage, the extracted nested entities are classified. This method effectively utilizes the boundary information of entities and partially matching spans during the training process through these two stages.

The span-based approach can also be implemented using pointer networks. Pointer networks can generate variable-length output sequences, breaking the limitation of fixed sequence length in general sequence-to-sequence models. In entity recognition models based on pointer networks, the text is used as input data. The model first predicts the start and end boundaries of entities to obtain entity spans, and then computes the representation of the entity spans to predict the entity types corresponding to the entity boundaries. Zhai et al. [23] introduced pointer networks in their model to perform sequence chunking and labeling. Li et al. [24] used GRU neural networks at the feature extraction layer to extract semantic features of the data and used pointer networks to eliminate the ambiguity of entity boundaries. Pointer networks have shown significant effectiveness in handling nested entities. However, in general, pointer networks treat multi-entity extraction tasks as multiple binary classification problems, which may lead to slow convergence when dealing with long sequences. Additionally, adding effective features or incorporating multi-task learning mechanisms can improve the recognition ability of span-based models. Zheng et al. [25] first extract all possible entity boundaries and then perform boundary detection and entity classification simultaneously. Eberts et al. [26] utilize a pre-trained Transformer model to jointly perform entity recognition and relation extraction in the span-based approach, and experiments show that pre-trained Transformer modules can enhance model performance. Yuan et al. [27] proposes a three-affine attention mechanism that incorporates label information into the span-based approach, using the three-affine mechanism to calculate entity scores for recognizing nested entities. This model interacts with both the span boundaries and the entity boundaries, and extends the two-affine mechanism with an additional dimension to obtain the three-affine mechanism. The span boundaries and the embeddings of the spans are used as the query vectors $Q$, while the word embeddings of each word within the span are used as the key vectors $K$ and value vectors $V$ in the attention mechanism. Compared to the general attention mechanism, this approach facilitates higher-dimensional interactions between $Q$ and $K$. Although this method handles the internal information within spans, the three-affine mechanism introduces an additional dimension compared to the two-affine mechanism, which may result in increased computational overhead when processing entity classification.

In summary, the previous two methods, namely the sequence labeling-based method and the hypergraph-based method, are not sensitive to entity boundary information. These methods may exhibit boundary ambiguity when dealing with nested entities, which affects the extraction of entity boundaries. To address these issues, this paper adopts the span-based method, which processes each potential entity boundary to extract all nested entities. Su et al. [28] proposed a nested entity recognition module called GlobalPointer in 2022. This module enumerates all potential entity boundaries of the text using upper triangular matrices. Each category has such an upper triangular matrix. The module calculates these matrices to obtain scores for each entity boundary in each category, thereby recognizing all nested entities in the text. However, a single GlobalPointer module may struggle to extract sufficient semantic information to distinguish similar categories in fine-

grained entity recognition datasets. Therefore, this paper constructs a deep semantic extraction module called ATT-IDCNNLR-ATTR, which leverages multiple layers of convolution, residual structures, and attention mechanisms to handle complex semantic information in the text and enable the model to extract more semantic information. Finally, this paper proposes a Chinese nested entity recognition model called BERT-ATT-IDCNNLR-ATTR-GlobalPointer, which can extract sufficient semantic information from the text, effectively distinguish similar fine-grained categories, and predict all nested entities and their corresponding entity types.

The innovations of this paper are as follows:

1. Replacing the classic conditional random field (CRF) of the traditional model with the GlobalPointer module, which is more suitable for nested entity recognition;
2. Using the classic IDCNN neural network module as the feature extraction layer of the model, optimizing the internal structure of IDCNN, incorporating attention mechanisms at multiple positions, and using residual structures to fuse various features, ultimately constructing a deep semantic extraction module called ATT-IDCNNLR-ATTR;
3. Creating multiple module combinations for each innovation point of the model and conducting multiple ablation experiments. In total, 20 ablation experiments were performed.

## 3. Model

The paper constructs the BERT-ATT-IDCNNLR-ATTR-GlobalPointer Chinese nested entity recognition model. In this model, BERT model is used as the word embedding layer, and data denoising is realized by multi-position multi-head attention mechanism. In order to improve the semantic extraction ability of the model, the paper constructs a deep semantic extraction module IDCNNLR, which extracts features through multi-layer convolution and fuses features of different levels to obtain richer features. In this model, position information is added to feature vectors by rotating position coding. The model uses the GlobalPointer module as the output layer to predict nested entities by calculating the entity boundary matrix. The overall frame diagram of the model in the paper is shown in Figure 1. The round block represents the entity predicted by the model, the rectangle represents the input text, and the rounded rectangle represents the module. In Figure 1, "北京大学… "(" Peking University.") is the input text instance of the model.

### 3.1. BERT

The first step in entity recognition is to convert the text into word vectors that can be used for numerical computations. In deep learning, there are generally two main methods for converting word vectors, namely word embedding layers and the BERT model. Both methods extract the semantic information of a word based on its surrounding context. For example, in the text "北方有许多城市，如北京市、石家庄市和哈尔滨市" ("There are many cities in the north, such as Beijing, Shijiazhuang, and Harbin"), the word "石家庄市" ("Shijiazhuang City") can be inferred as a word related to cities based on the surrounding words "北京" ("Beijing City") and "哈尔滨市" ("Harbin City"). To enhance the ability to extract semantic information of a specific word from its surrounding context, the BERT model uses a masked language model task for pre-training. The masked language model task randomly masks some vocabulary, and the model predicts the masked words by considering the context of the masked words, thereby improving the model's ability to extract semantic information from each position. Sometimes, relying solely on the surrounding words may be insufficient to extract the true semantic information of a specific word. For example, in the two texts "You came too early today" and "You came too late today", the context of the words "early" and "late" is exactly the same, but their semantics are completely opposite. In such cases, the word embedding layer, which relies on the surrounding words to extract the semantic information of a specific word, may struggle to capture the correct semantics of the specific word. To address this issue, the BERT model incorporates the next sentence prediction task to enhance the model's perception of semantic relationships between sentences. The next sentence prediction task selects one sentence as

the previous sentence and randomly selects the next sentence of that sentence as the next sentence with a probability of 50%. The model predicts whether these two sentences are consecutive. Through the next sentence prediction task, the model can better perceive the semantic connections between sentences. In summary, this paper adopts the BERT model as the word embedding layer of the proposed model, leveraging the BERT model to extract more semantic information from the text and improve the quality of word vectors. The pre-training file name used for the BERT model in this paper is bert-base-chinese.

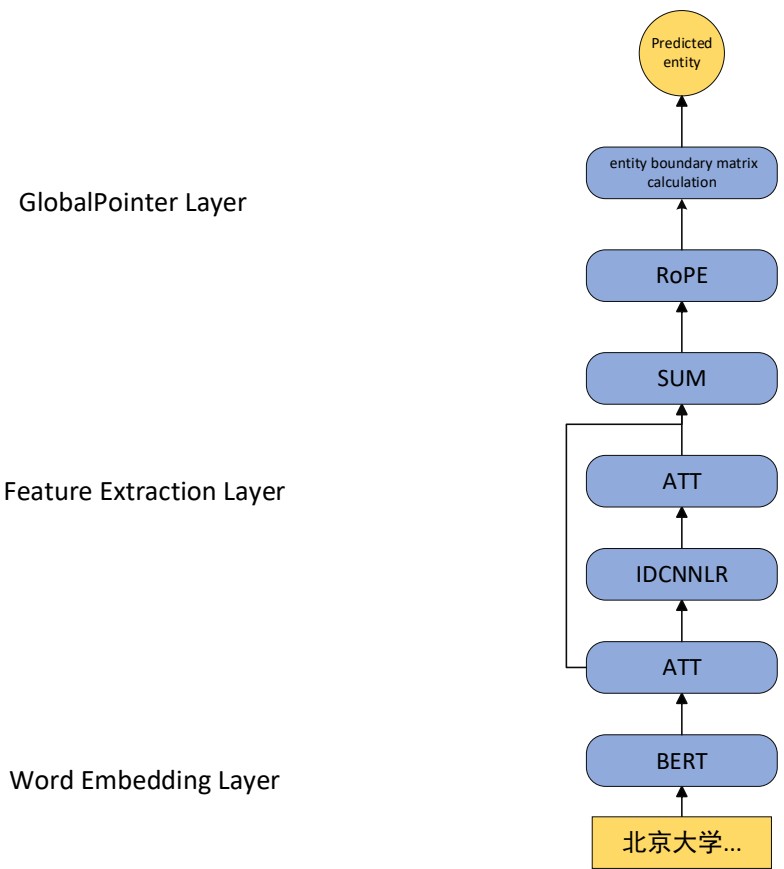

**Figure 1.** Overall architecture of BERT-ATT-IDCNNLR-ATTR-GlobalPointer.

### *3.2. Multi-Head Self-Attention Mechanism*

Attention mechanism is a method that emphasizes important information in data. By using the attention mechanism, the model can enhance its perception of important information and extract more semantic information. For example, in the task of entity recognition, in the text "Peking University is located in Haidian District, Beijing", the word "in" is a preposition that often does not carry much actual meaning. The model can utilize the attention mechanism to identify that "in" is likely not an important character for entity recognition and assign it a lower weight during computation, thereby weakening the contribution of non-important information. The attention mechanism highlights the semantic information of important characters, improving the quality of data features.

In deep learning, attention mechanisms can be categorized into three types: global attention mechanism, local attention mechanism, and self-attention mechanism. A character in the text has a certain semantic correlation with other characters, and this character can represent more semantic information of the text. The self-attention mechanism improves the weight of characters that have strong correlations with other characters, thereby increasing the semantic information of features. The attention score in the self-attention mechanism is obtained through matrix multiplication between key vectors and query vectors, where the attention score represents the similarity between each position element of

the key vector and the query vector. The matrix product of the query vector and the key vector is matrix multiplication. The higher the similarity between a character and other characters, the stronger the semantic correlation the character has with all the characters in the text. As a result, the character can express more semantic information of the text, and it will have a larger value in the attention score matrix, indicating the character has higher weight and more semantic information in the feature vector. For example, in the text "中国有56个民族，包括汉族、满族和回族" ("There are 56 ethnic groups in China, including Han, Manchu and Hui"), there are four entities representing ethnic groups: "汉族" ("Han nationality"), "满族" ("Manchu"), "回族" ("Hui nationality") and "民族" ("nation"). The character "族" appears in all four entities, indicating a high probability of being an entity character. The attention mechanism assigns a higher weight to the character "族", increasing the model's focus on this character. As a result, the model can recognize more entities that contain the character "族" and represent ethnic groups, such as "维吾尔族" ("Uighur nationality") and "苗族" ("Miao nationality").

In summary, the attention mechanism enhances the quality of data features by emphasizing important information. It can be categorized into a global attention mechanism, local attention mechanism, and self-attention mechanism. Unlike the other two attention mechanisms, the self-attention mechanism improves the weight of characters with strong correlations, allowing the model to extract more semantic information.

The self-attention mechanism calculates attention scores based on the matrix computation of feature vectors themselves, providing global attention scores for the text. However, the single-head self-attention mechanism has limited dimensions in processing data, leading to insufficient extraction of semantic information. Liao et al. [29] introduced the multi-head self-attention mechanism to solve the problem that military named entity recognition requires a lot of domain specific knowledge. Therefore, in this paper, the single-head self-attention mechanism is replaced with the multi-head self-attention mechanism to enhance the perception of semantic connections between characters. The multi-head self-attention mechanism changes the dimensions of key, value, and query vectors compared to the single-head self-attention mechanism. The last dimension of the feature vectors is divided into *head_num* parts, adjusting the feature vector dimension from (*batch_size, seq_size, word_embedding_size*) to (*batch_size, seq_size, word_embedding_size/8*). This mechanism splits a feature vector into multiple groups on the corresponding dimension of the word vectors, perceiving the semantic connections between characters from more dimensions. *batch_size* is the batch size during training, *seq_len* is the text sequence length, and *word_embedding_size* is the dimension of the word vector.

The workflow of the multi-head attention mechanism is as follows. Firstly, the feature vectors undergo dimension transformation to obtain multiple sets of feature vectors. In this paper, the key $k$, value $v$, and query vectors $q$ in the attention mechanism are all feature vectors from the input. The key $k$, value $v$, and query vectors $q$ pass through their respective fully connected layers to obtain three new feature vectors: $q_{all}$, $k_{all}$, and $v_{all}$. These three new feature vectors undergo internal computations within the attention mechanism to obtain the output vectors $h_i$ for each attention head. These attention head outputs $h_i$ are concatenated in the last dimension in sequence to obtain the overall output vector $h_{all}$. Finally, the output vector goes through fully connected layers, dropout layers, and layer normalization to obtain the final output vector $h_{last}$. The calculation formulas are shown in Equations (1) and (2). The internal structure of the multi-head attention mechanism is shown in Figure 2. In Figure 2, the yellow circular icon represents the feature vector in data transmission, and the blue rounded rectangle represents the feature processing layer.

$$h_i = f\left(W_i^{(q)}q, W_i^{(k)}k, W_i^{(v)}v\right) \tag{1}$$

$$h_{last} = W_{last}[h_1, h_2 \ldots h_n] \tag{2}$$

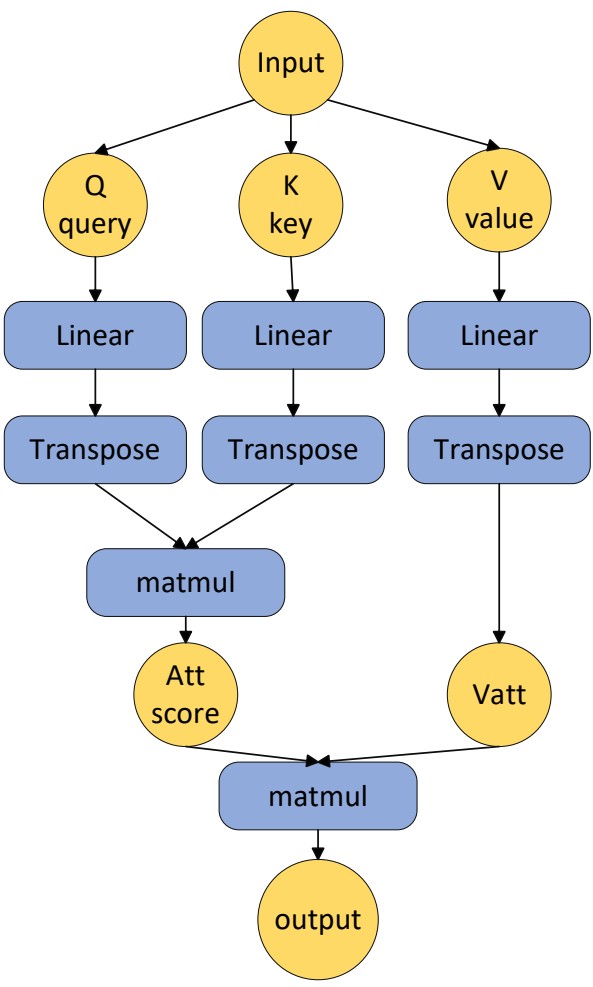

**Figure 2.** The internal structure of the multi-head attention mechanism.

*3.3. IDCNNLR*

There has been numerous research on entity recognition based on Bidirectional LSTM (BiLSTM), which can extract more sequential information to capture the semantic relationships between words. Wei et al. [30] used BILSTM to extract the time sequence information of texts and built a BERT-BILSTM-CRF entity recognition model to realize entity recognition of texts in educational emergencies. Yang et al. [31] add BILSTM and Transformer to the entity recognition model. These two modules improve the semantic extraction capability of the feature extraction layer. However, BiLSTM cannot fully utilize the computational power brought by GPU parallelism, thus being unable to significantly improve the model performance. Each step of the CNN calculation is relatively independent for the GPU, and each step of the CNN calculation can be carried out at the same time, so as to achieve parallel computing. Because of the different calculation methods, LSTM and other modules do not have this feature. CNN has also been widely applied in entity recognition research, as it possesses strong capabilities in extracting local features to capture the semantic relationships between words. Since CNN can only extract local semantic information, many studies increase the depth of the CNN network to extract broader semantic information, which may lead to overfitting issues. Pooling layers have been used to aggregate text information from multiple convolution scales, which can lead to the loss of necessary information. To address these challenges, Strubell et al. [32] proposed an Induced Dilated Convolutional Neural Network (IDCNN). IDCNN replaces the ordinary CNN with dilated convolutions, enlarging the receptive field of the convolution kernels without changing the number of model parameters. This allows the convolutional layers to learn global semantic information from the text.

The internal structure of the IDCNN model is four layers of DCNN, where each DCNN layer contains two standard convolutions and one dilated convolution with a dilation value of 2. Before each convolutional layer, the LeakyReLU function is applied to ensure that the input data remains within an appropriate range. Following the activation function layer, a Layer Normalization (LayNorm) layer is added to prevent the gradient vanishing problem after several convolutional layers. The internal structure of DCNN is shown in Figure 3. The convolutional layers of DCNN are one-dimensional, and the data is propagated forward through four layers of DCNN sequentially. IDCNN swaps the second and third dimensions of the input data and then swaps them again to obtain the output vector.

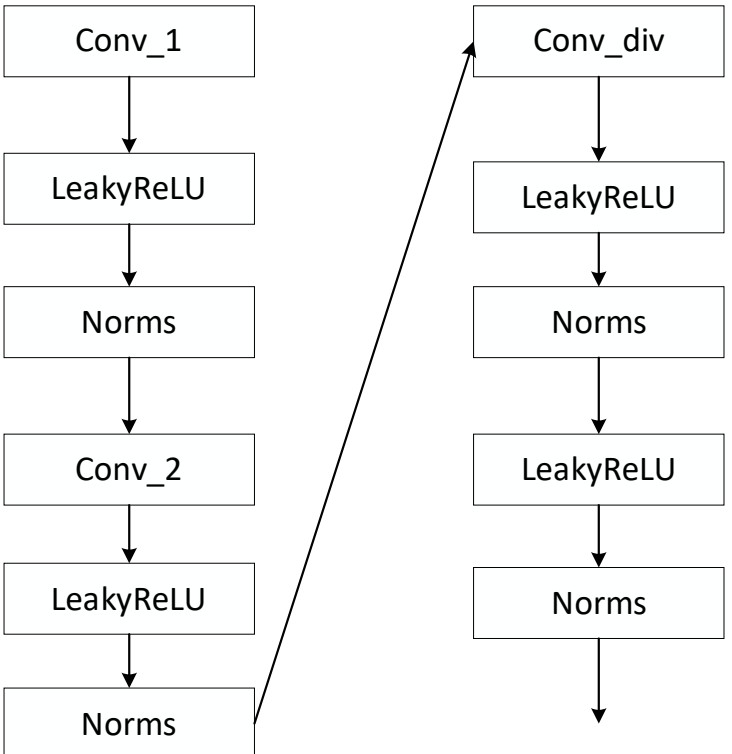

**Figure 3.** Internal structure of DCNN.

The paper improved the internal structure of the classical module IDCNN and obtained IDCNNLR module. Compared with IDCNN module, the extraction capability of IDCNNLR module has been enhanced to some extent. The data is sequentially propagated forward through four layers of DCNN of the IDCNN model, and these layers extract text features at different depths. IDCNNLR module does residuals summation of the output features for four layers of DCNN, and obtains the fusion features at different text depths. The information contained in features at different depths is non-overlapping, and fusing features from different depths provides the model with more semantic information. The internal structure of IDCNNLR is shown in Figure 4.

The ReLU activation function can address the gradient vanishing problem in LSTM. It sets the gradient of negative input to 0, preventing the update of these parameters, and sets the gradient of positive input to 1, allowing the update of data gradients, ensuring that the model parameters are not difficult to update with low gradients. However, ReLU sets the gradient of negative values to 0, making it difficult for negative values to be updated, resulting in suboptimal training performance. To address this issue, IDCNNLR replaces the ReLU activation function with the LeakyReLU activation function. The LeakyReLU function is based on the ReLU activation function but changes the gradient of negative values to a positive parameter, typically set to 0.01. Through ablation experiments in subsequent

chapters, it is concluded that replacing the ReLU function with the LeakyReLU function in the model leads to improved performance.

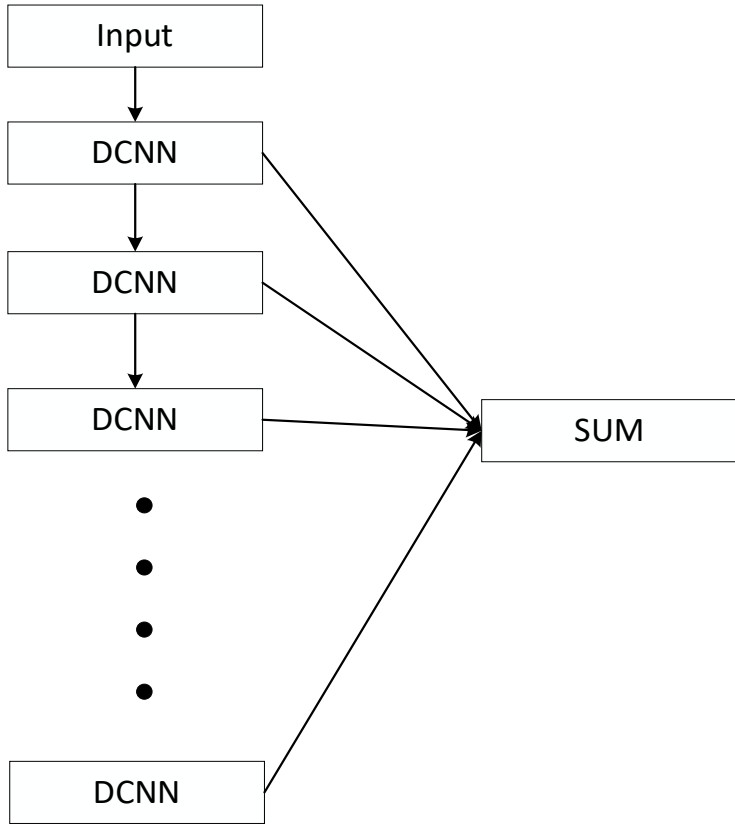

**Figure 4.** Internal structure diagram of IDCNNLR.

### 3.4. GlobalPointer

Research on flat entity recognition has been extensive, and many flat entity recognition models have achieved high performance on multiple flat entity recognition datasets. Yin et al. [33] proposed a Multi-criteria Fusion Model, which has achieved excellent experimental results on the flat entity recognition dataset MSRA. Cao et al. [34] constructed a Bert-MRC-Biaffine entity recognition model, which has achieved excellent experimental results on the flat entity recognition dataset CCKS2017. These flat entity recognition datasets are characterized by the fact that each text fragment can represent only one entity, and entities cannot share the same characters. However, in a complex context, one entity may contain another entity, which is impossible in a flat entity recognition dataset. It is necessary to evaluate the capability of entity recognition models with nested entity recognition datasets. In real life, nested entities are quite common. For example, in the text segment "北京大学" ("Peking University"), the text segment "北京大学" is an organizational entity, and the text segment "北京" ("Beijing, China") is a location entity. These two entities overlap in the text segment "北京大学". The annotation method based on individual character labels alone cannot identify multiple overlapping entities. To address this issue, Su et al. proposed a nested entity recognition model called GlobalPointer, which considers all entity boundaries in the text. The model constructs a score matrix for entity boundaries to calculate the scores of entity boundaries for different categories, predicting the nested entities and their types in the text.

The GlobalPointer module first splits the input feature vectors into two equally sized feature vectors $q$ and $k$ in the last dimension and adds positional encoding information using RoPE [35]. The two feature vectors $q$ and $k$ undergo multiple steps of multidimensional matrix multiplication to obtain the raw score matrix of entity boundaries $Logits_{first}$.

The module uses masking matrix to discard the entity boundary that exceeds the length of the sequence, eliminates invalid entity boundary from the lower triangular matrix, and finally obtains the score matrix of entity boundary $Logits_{final}$. The score matrix contains confidence scores for each entity boundary in each category, and the model predicts nested entities and their entity types based on these confidence scores.

The RoPE (Rotation Positional Encoding) method incorporates positional information into the feature vectors. The working principle of RoPE is to first calculate the position embedding vectors *pos_emb* based on the size of the feature vectors $q$ and $k$. The odd-indexed and even-indexed data of the embedding vectors *pos_emb* are separately extracted, and each element is duplicated in its original position, resulting in sine positional vectors $pos_{sin}$ and cosine embedding vectors $pos_{cos}$. The feature vectors $q$ and $k$ are split into odd-indexed and even-indexed data, respectively, which are then concatenated to obtain new feature vectors $q_2$ and $k_2$. The original eigenvector takes the matrix product of the sinusoidal embedding vector $pos_{sin}$, the new eigenvector takes the matrix product of the cosine embedding vector $pos_{cos}$, and the sum of the two product results is the final eigenvector containing the sum of the position information $q_w$ and $k_w$. The calculation formulas of $q_w$ and $k_w$ are shown in Equations (3) and (4).

$$q_w = q_w \times pos_{cos} + q_2 \times pos_{sin} \tag{3}$$

$$k_w = k_w \times pos_{cos} + k_2 \times pos_{sin} \tag{4}$$

## 4. Experiment

### 4.1. Datasets

This paper validates the effectiveness of the overall model and each module on two Chinese fine-grained entity recognition datasets: CMeEE dataset and CLUENER2020 dataset.

The CMeEE dataset [36] is a nested entity recognition dataset for Chinese medical texts, which categorizes medical named entities into nine classes, including diseases (dis), symptoms (sym), drugs (dru), medical equipment (equ), medical procedures (pro), body parts (bod), medical examination items (ite), microorganisms (mic), and departments (dep). This dataset allows nested entities of different categories within an entity. For example, in the text "pediatric pleural diseases", there are two entities: "pleural diseases" and "pediatric pleural diseases". The entity "pleural diseases" is nested within the entity "pediatric pleural diseases". Compared to flat entity recognition datasets, the CMeEE dataset can evaluate the model's ability to recognize nested entities. The information statistics of the CMeEE dataset are listed in Table 1.

**Table 1.** Statistics of the datasets.

| Dataset | Train | Dev | Test | Classes | Total Number |
|---------|-------|-----|------|---------|--------------|
| CMeEE | 15,000 | 5000 | Null | 9 | 20,000 |
| CLUENER2020 | 10,748 | 1343 | Null | 10 | 12,091 |

The download link of CMeEE dataset is "https://tianchi.aliyun.com/dataset/95414". The date of access to the website is 30 December 2023. Under this link, the dataset is named "CMeEE-V2".

CLUENER2020 [37] is a Chinese fine-grained entity recognition dataset, which is obtained by annotating part of the data of THUCTC open source text classification dataset of Tsinghua University. CLUENER2020 includes a total of 10 tag types, including positions, names, ads, companies, governments, titles, games, movies, organizations/institutions, attractions, and more. The entities of this dataset are labeled using 10 fine-grained entity categories. Each entity category is evenly distributed in the data set. The text in the data set is long text. The information statistics of the CLUENER2020 dataset are shown in Table 1.

The download link of CLUENER2020 dataset is "https://tianchi.aliyun.com/dataset/144362?spm=a2c22.28136470.0.0.d3554a0a9cI392&from=search-list". The date of access to the website is 30 December 2023. Under this link, the dataset is named "cluener_public".

### 4.2. Experimental Metrics

To really evaluate the performance of the model and compare the effects of different models. This paper uses three metrics, *P* (precision), *R* (recall), and *F* (*F1* score), to evaluate the performance of the model. Among them, *P* represents the ratio of correctly identified entities to all identified entities, *R* represents the ratio of correctly identified entities to all entities that should be identified, and *F* is a comprehensive evaluation index that combines *P* and *R*. The formulas for calculating the three metrics are shown in Equations (5)–(7).

$$P = TP/(TP + FP) \times 100\% \tag{5}$$

$$R = TF/(TP + FN) \times 100\% \tag{6}$$

$$F = 2 \times P \times R/(P + R) \times 100\% \tag{7}$$

In the above equations, *TP* represents the number of samples that are actually positive and predicted as positive, *FP* represents the number of samples that are actually negative but predicted as positive, and *FN* represents the number of samples that are actually positive but predicted as negative.

### 4.3. Experimental Environment

This paper uses Python 3.8.16 and PyTorch 1.12.0+cu116 as the configuration environment for the experiments. BERT pre-trained language model is used to generate word vectors. Table 2 shows the settings of the hyperparameters used in the experiments. The input of the model in this paper is text data, and the input size depends on the text length. The size of the input data is (4, *seq_len*). In this model, the attention mechanism and IDCNN do not change the size of the data, and the output size of these two modules is (4, *seq_len*, 768). In DCNN, the size of the convolution kernel is 3 × 3. This value *d_classes* represents the number of categories for every dataset. This value *seq_len* is the length of the text sequence.

**Table 2.** Setting of the hyperparameters used in the experiment.

| Parameter | Value |
|---|---|
| BILSTM hidden size | $128 \times d\_classes$ |
| Number of BILSTM layers | 1 |
| Dropout rate | 0.1 |
| Optimizer | Adam |
| Learning rate | $7 \times 10^{-5}$ |
| Epoch | 20 |
| Batch size | 4 |
| Bert output dimension | 768 |
| BERT | Bert-base-chinese |
| Number of IDCNN layers | 12 |
| Number of model parameters | 177,867,080 |

The training mode of the model in this paper is as follows. The model is trained on the training set for 20 rounds, and the experimental effect is verified on the verification set after each training round. After the training, the experimental effect with the best effect on the verification set is selected as the final experimental result, and the model parameters of the effect are the optimal model parameters. The loss function used in model training is multi-label classification cross entropy. The loss function calculates the loss values for each class, sums these loss values and averages them.

The equipment environment of this experiment is described as follows. The computer type is "BATTLE-AX Z790M-PLUS D5", the processor type is "13th Gen Intel(R) Core(TM) i7-13700KF 3.40GHz", and the operating system is a 64-bit operating system. GPU device is "NVIDIA GeForce RTX 4090 (24G)".

### 4.4. Comparative Experiment

The comparative experimental results of the model on the CMeEE dataset are shown in Table 3, and the comparative experimental results on the CLUENER2020 dataset are shown in Table 4. The analysis of the comparative experiments is as follows. In Tables 3 and 4, the bolded data represent the best experimental results for this metric.

**Table 3.** Contrast experiments of CMeEE dataset.

| Compare Models | P% | R% | F1% |
|---|---|---|---|
| BERT-ATT-IDCNNLR-ATTR-GlobalPointer | **71.276** | **69.127** | **69.617** |
| BERT-GlobalPointer | 69.405 | 69.032 | 68.605 |
| FLR-MRC | 66.79 | 66.25 | 66.52 |
| GPa +RoBERTa-large+SoftLexicon | NULL | NULL | 68.13 |
| TPORE | 63.73 | 66.25 | 64.94 |

**Table 4.** Contrast experiments of CLUENER2020 dataset.

| Compare Models | P% | R% | F1% |
|---|---|---|---|
| BERT-ATT-IDCNNLR-ATTR-GlobalPointer | **77.325** | **81.407** | **79.285** |
| BERT-GlobalPointer | 77.034 | 78.519 | 77.906 |
| BERT-CRF | 71.867 | 77.669 | 74.655 |
| BERT-BILSTM-CRF | 73.978 | 75.423 | 74.693 |
| BERT-ATT-BILSTM-CRF | 74.525 | 74.186 | 74.355 |
| BERT-IDCNN-CRF | 74.244 | 78.352 | 76.243 |

Comparing BERT-ATT-BILTAR-GlobalPointer with BERT-GlobalPointer, the addition of IDCNNLR and multi-position attention mechanism in the BERT-GlobalPointer model improves the *F1* score of the model by 1.012% on the CMeEE dataset and by 1.378% on the CLUENER2020 dataset. The improved performance indicates that the IDCNNLR module can extract deeper semantic information, and the multi-position attention mechanism can highlight important information, thereby enhancing the model's performance.

In the comparative experiments on the CMeEE dataset, compared to the FLR-MRC [38] model, the *F1* score of our model improved by 3.097%. Compared to the GPa+RoBERTa-large+SoftLexicon [39] model, the *F1* score of our model improved by 1.487%. Compared to TPORE, the *F1* score improved by 4.677%. These experiments demonstrate that combining GlobalPointer with the IDCNNLR deep semantic feature extraction layer and the multi-position attention mechanism can achieve excellent results, surpassing the current baseline models.

In the comparative experiments on the CLUENER2020 dataset, our model outperformed the BERT-CRF model by improving the *F1* score by 4.629%. Compared to the BERT-IDCNN-CRF model, our model improved the *F1* score of our model by 3.042%. When compared to the BERT-BILSTM-CRF model, our model improved the *F1* score by 4.591%. Furthermore, compared to the BERT-ATT-BILSTM-CRF model, our model improved the *F1* score by 4.929%. These experiments demonstrate that the BERT-ATT-IDCNNLR-ATTR-GlobalPointer model performs better than classical entity recognition neural network models.

### 4.5. Ablation Experiments

The ablation experiment in this study was divided into six ablation experiment modules. The six modules are the multi-head attention mechanism ATT, IDCNN residual structure, IDCNN LeakyReLU activation function, IDCNNLR, and rotary position coding REPO. In this study, the control variable method was used for experimental design. In order to prove the effectiveness of the six modules, the study adopted a unified data processing method, the same operating environment and training parameter Settings. The differences between the models only exist in the different structural components, and the common parts are consistent. When the model is trained on the same data set, the hyperparameters of the model are the same. A total of 20 module combinations were tested in ablation experiments. Table 5 shows the ablation test results of each module. In Table 5, the bolded data represent the best experimental results for this metric. The following is the introduction and analysis of each ablation experiment:

**Table 5.** Twenty ablation experiments of the present model.

| Compare Models | CMeEE | | | CLUENER020 | | |
|---|---|---|---|---|---|---|
| | P% | R% | *F1%* | P% | R% | *F1%* |
| BERT-ATT-IDCNNLR-ATTR-GlobalPointer | **71.276** | 69.127 | **69.617** | 77.325 | **81.407** | **79.285** |
| BERT-ATT-IDCNNR-ATTR-GlobalPointer | 66.944 | **71.577** | 68.654 | 78.218 | 79.836 | 78.987 |
| BERT-IDCNNLR-GlobalPointer | 70.530 | 68.634 | 68.872 | 74.338 | 77.964 | 76.050 |
| BERT-ATT-IDCNNLR-GlobalPointer | 69.385 | 69.162 | 68.639 | 71.790 | 80.585 | 75.897 |
| BERT-ATT-IDCNNR-GlobalPointer | 68.223 | 69.677 | 68.355 | 73.138 | 76.036 | 73.635 |
| BERT-ATT-GlobalPointer | 70.079 | 68.952 | 68.844 | 76.894 | 78.997 | 77.906 |
| BERT-ATT-IDCNN-ATTR-GlobalPointer | 68.558 | 71.046 | 69.240 | 77.643 | 80.852 | 79.169 |
| BERT-GlobalPointer | 69.405 | 69.032 | 68.605 | 77.034 | 78.519 | 77.736 |
| BERT-ATT-GlobalPointer | 68.692 | 69.586 | 68.634 | 76.434 | 76.733 | 75.910 |
| BERT-ATT-IDCNNL-ATTR-GlobalPointer | 69.942 | 68.097 | 68.386 | 77.613 | 80.088 | 78.801 |
| BERT-ATT-IDCNNL-GlobalPointer | NULL | NULL | NULL | 77.397 | 81.070 | 79.156 |
| BERT-IDCNNR-GlobalPointer | 70.786 | 67.673 | 68.538 | 73.806 | 78.574 | 76.056 |
| BERT-IDCNNL-ATTR-GlobalPointer | 68.796 | 69.265 | 68.498 | 78.307 | 79.955 | 79.087 |
| BERT-IDCNNL-GlobalPointer | 69.635 | 69.938 | 69.255 | **80.247** | 77.794 | 78.964 |
| BERT-ATT-IDCNNR-GlobalPointer | 68.223 | 69.677 | 68.355 | 72.983 | 79.063 | 75.851 |
| BERT-IDCNNLR-ATTR-GlobalPointer | 70.059 | 68.638 | 68.775 | 78.042 | 80.619 | 79.251 |
| BERT-IDCNNR-ATTR-GlobalPointer | 67.613 | 71.822 | 69.143 | 75.558 | 75.104 | 74.714 |
| BERT-ATT-IDCNNLR-ATTR-GlobalPointer(without RePO) | 68.381 | 66.853 | 66.953 | 77.587 | 80.533 | 78.986 |
| BERT-IDCNNLR-GlobalPointer(without RePO) | 69.667 | 66.246 | 67.261 | 73.300 | 78.271 | 75.621 |
| BERT-GlobalPointer(without RePO) | 69.852 | 65.694 | 67.029 | 78.123 | 78.695 | 78.379 |

#### 4.5.1. Multi-Head Attention Mechanism

In the comparison between BERT-ATT-IDCNNLR-ATTR-GlobalPointer and BERT-IDCNNLR-GlobalPointer, on the CMeEE dataset, the model achieved a 0.745% improvement in *F1* score, while on the CLUENER2020 dataset, the model achieved a 3.234% improvement in *F1* score.

In the comparison between BERT-ATT-IDCNNLR-ATTR-GlobalPointer and BERT-ATT-IDCNNLR-GlobalPointer, on the CMeEE dataset, the model achieved a 0.978% improvement in *F1* score, while on the CLUENER2020 dataset, the model achieved a 3.387% improvement in *F1* score.

The experiments indicate that the multi-head attention mechanism improves the performance of the model when placed at different positions in various combination models. The

multi-head attention mechanism performs noise reduction on the data, highlighting important information in the features, and enhancing the model's recognition performance.

### 4.5.2. Residual Structure of IDCNNLR

In the comparison between BERT-ATT-IDCNNLR-ATTR-GlobalPointer and BERT-ATT-IDCNNL-ATTR-GlobalPointer, on the CMeEE dataset, the model achieved a 1.231% improvement in *F1* score. On the CLUENER2020 dataset, the model achieved a 0.483% improvement in *F1* score.

In the comparison between BERT-IDCNNLR-ATTR-GlobalPointer and BERT-IDCNNL-ATTR-GlobalPointer, on the CMeEE dataset, the model achieved a 0.277% improvement in *F1* score. On the CLUENER2020 dataset, the model achieved a 0.164% improvement in *F1* score.

The experiments indicate that incorporating the residual structure into the IDCNN module improves the performance of multiple combination models. The residual structure combines feature vectors from different depths of the DCNN module, creating a multi-scale one-dimensional convolution structure that extracts more semantic features from the text.

### 4.5.3. LeakyReLU activation function

In the comparison between BERT-ATT-IDCNNLR-ATTR-GlobalPointer and BERT-ATT-IDCNNR-ATTR-GlobalPointer, the model achieved a 0.963% improvement in *F1* score on the CMeEE dataset and a 0.298% improvement on the CLUENER2020 dataset.

In the comparison between BERT-ATT-IDCNNLR-GlobalPointer and BERT-ATT-IDCNNR-GlobalPointer, the model achieved a 0.283% improvement in *F1* score on the CMeEE dataset and a 0.046% improvement on the CLUENER2020 dataset.

In the comparison between BERT-IDCNNLR-GlobalPointer and BERT-IDCNNR-GlobalPointer, the model achieved a 0.334% improvement in *F1* score on the CMeEE dataset, but there was no improvement on the CLUENER2020 dataset.

In the comparison between BERT-IDCNNLR-ATTR-GlobalPointer and BERT-IDCNNR-ATTR-GlobalPointer, the model did not improve in *F1* score on the CMeEE dataset, but it achieved a 4.537% improvement in *F1* score on the CLUENER2020 dataset.

The experiments demonstrate that changing the activation function of IDCNN from ReLU to LeakyReLU improves the performance of multiple combination models. LeakyReLU addresses the dying ReLU problem by allowing negative values to also obtain gradients, thereby enhancing the training effectiveness of the model.

### 4.5.4. IDCNNLR

In the comparison between BERT-ATT-IDCNNLR-ATTR-GlobalPointer and BERT-ATT-GlobalPointer, the model achieved a 0.983% improvement in *F1* score on the CMeEE dataset and a 3.375% improvement on the CLUENER2020 dataset.

In the comparison between BERT-IDCNNLR-GlobalPointer and BERT-GlobalPointer, the model achieved a 0.266% improvement in *F1* score on the CMeEE dataset, but there was no improvement within 20 rounds on the CLUENER2020 dataset.

In the comparison between BERT-IDCNNLR-ATTR-GlobalPointer and BERT-ATT-GlobalPointer, the model achieved a 0.141% improvement in *F1* score on the CMeEE dataset and a 1.344% improvement on the CLUENER2020 dataset.

The experiments demonstrate that incorporating the IDCNNLR module into multiple combination models leads to performance improvements. The IDCNNLR module, as a feature extraction layer, enhances the model's ability to extract semantics and explore deep semantic connections between nested entities, thereby improving the model's recognition capability.

### 4.5.5. Rotate Position Encoding RePO

In the comparison between BERT-ATT-IDCNNLR-ATTR-GlobalPointer and BERT-ATT-IDCNNLR-ATTR-GlobalPointer (without RePO), the model achieved a 2.664% improvement in *F1* score on the CMeEE dataset and a 0.298% improvement on the CLUENER2020 dataset.

In the comparison between BERT-IDCNNLR-GlobalPointer and BERT-IDCNNLR-GlobalPointer (without RePO), the model achieved a 1.519% improvement in *F1* score on the CMeEE dataset and a 0.429% improvement on the CLUENER2020 dataset.

In the comparison between BERT-GlobalPointer and BERT-GlobalPointer (without RePO), the model achieved a 0.934% improvement in *F1* score on the CMeEE dataset, but there was no improvement on the CLUENER2020 dataset.

The experiments demonstrate that incorporating the rotated position encoding (RePO) into the GlobalPointer module leads to performance improvements in multiple combination models. RePO is able to capture the positional relationships between different words and embeds positional information into the features.

### 4.5.6. Summary and Analysis of Ablation Experiment

It should be noted that the performance of some combination models may be lower than that of the base model. This is because the IDCNN module has a large number of model parameters, which requires high training costs. The addition of sub-modules may slow down the training speed of the model, making it difficult for the model to achieve the expected performance within 10 or 20 rounds. The CMeEE dataset contains a large number of nested entities and focuses on evaluating the model's ability to recognize nested entities. The content of this dataset is mainly related to medical text information. The entity categories in this dataset are medical entity categories, which are more fine-grained compared to fine-grained entity recognition datasets in daily texts. The CLUENER2020 dataset is a dataset of daily texts, aiming to cover most of the information in daily texts. The entity categories in this dataset are broader compared to the CMeEE dataset. In summary, the two datasets have different focuses, and the performance improvement of the model in the same ablation experiment may vary on these two datasets.

In conclusion, in the above ablation experiments, the sub-modules of our model have all contributed to the improvement of model performance. Our model outperforms the baseline model on both Chinese entity recognition datasets.

## 5. Conclusions and Future Work

In real-life scenarios, text context can be complex, and texts such as medical information and daily expression often contain a large number of nested entities. The task of nested entity recognition in complex text is an important challenge. Improving the model's ability to recognize nested entities can enhance the processing speed of organizations, such as medical institutions, in handling large volumes of textual data. Enhancements to nested entity recognition models are also crucial for the development of knowledge graphs. The identified nested entities and entity categories can serve as knowledge for downstream tasks such as knowledge fusion and reasoning within the knowledge graph. Improving the entity recognition capability in each knowledge graph task can reduce error propagation from the entity recognition stage and improve the performance of each downstream task.

Currently, there is relatively limited research on nested entity recognition in Chinese, with most Chinese entity recognition studies focusing on flat entity recognition tasks. General flat entity recognition models, such as sequence labeling-based models, cannot handle the recognition of nested entities in text. To address this issue, our model adopts the GlobalPointer module to handle potential entity boundaries and extract nested entities from the text. Classical entity recognition models lack sufficient feature extraction capabilities to capture deep semantic relationships between nested entities. To tackle this problem, our model incorporates the BERT model as the word embedding layer to extract word-level semantic information. It also employs a combination of the multi-position multi-head attention mechanism and the semantic extraction module, IDCNNLR, to extract more semantic

information from the text. Experimental results demonstrate that the GlobalPointer module exhibits certain effectiveness in handling nested entities. With a combination of BERT layers, attention mechanisms, and IDCNNLR modules, our model can capture deeper semantic information between nested entities, thereby improving recognition performance. In summary, the future work of Chinese nested entity recognition is prospected as follows in this chapter.

In the future, further improvements to the IDCNNLR module can be made by incorporating more complex network structures. Stronger semantic extraction modules enable the model to capture deeper semantic features, thereby enhancing overall performance. The optimization of the GlobalPointer module can be explored to improve the model's ability to compute entity boundaries.

**Author Contributions:** Conceptualization, J.L.; Methods, J.L. and W.L.; Verification, J.L., Y.G. and J.G.; Formal analysis, X.Z.; Resources, J.L.; Data Management, Y.G.; Writing—Original manuscript preparation, J.L.; Writing—Review and editing, W.L. All authors have read and agreed to the published version of the manuscript.

**Funding:** This work was supported by the Ningxia Natural Science Foundation Project (2021AAC03215), the Basic Scientific Research in Central Universities of North Minzu University (2021JCYJ12), the National Natural Science Foundation of China (62066038, 61962001).

**Data Availability Statement:** The data presented in this study are available on request from the corresponding author. Because the data sets in this article were developed by individual institutions, the data are not publicly available.

**Conflicts of Interest:** The authors declare no conflict of interest.

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
