# Peer review of "Research on Chinese Nested Entity Recognition Based on IDCNNLR and GlobalPointer"

_asi, doi:10.3390/asi7010008_

Round 1
Reviewer 1 Report
Comments and Suggestions for Authors
Comments:
The article is well structured and presented. It has been felt strongly that it is useful to consider the nested or hierarchical structure within the text context. However, the followings need to be addressed to enhance the article:
In line 20, please omit the word ‘fully’.
In line 65, please mention some of the research. You could say numerous researchers with some notable references.
In line 66 and 67, some quantitative evidence or references are needed to clarify ‘remarkable improvement’. This is because of the major motivation of using Deep learning-based NER of this paper.
In line 77, which medical field? Is it in general or overall. Specific references could enhance the article.
In line 100, what is meant by ‘fine-grained’?
In line 185, please change the words ‘one study’.
In line 328, is it does not clear the use of matrix multiplication. Is it because of multiple layer? Or will the result dependent on the type of operation? In line 380, ‘numerous research’ with some references should be used instead of saying ‘abundant research’.
In line 382 and 383, it is not clear why BiLSTM cannot fully utilise the computational power brought by GPU parallelism.
As previously said, some references are needed for lines 428 to 431. Specifically, what datasets are used? How similar are these datasets compared to the datasets that this paper uses?
Lines 457 to 461, it is not clear about the eigenvectors. Why are they used? Is there any specific purpose? Are these eigenvectors being paramount features?
In line 465, how are the datasets fine-grained? Are they already cleaned and processed? Any additional features are imported into the datasets, i.e., linked with external data sources?
Author Response
The modification content is attached.

Reviewer 2 Report
Comments and Suggestions for Authors
The manuscript submitted to Applied System Innovation (asi-2754183) titled “Research on Chinese nested entity recognition based on IDCNNLR and GlobalPointer” is about a model that uses a GlobalPointer module to extract the deep semantic information that is dedicated to Nested Named Entity Recognition (NNER) for identification. This is done by building an Induced Dilated Convolutional Neural Network LeakyReLU (IDCNNLR) semantic extraction module. This allows the convolutional layers to learn global semantic information from the text instead of a CNN-LSTM model architecture. Results show that this modeling-simulating experiment “exhibits improvement compared to baseline models”. In addition, the contribution of performance of components to the overall system, by removing them in turn, shows effective performance enhancement.
This paper “Research on Chinese nested entity recognition based on IDCNNLR and GlobalPointer” is following the one in Applied Science that is “Li, W., Liu, J., Gao, Y., Zhang, X., & Gu, J. (2023). Chinese Fine-Grained Named Entity Recognition Based on BILTAR and GlobalPointer Modules. Applied Sciences, 13(23), 12845 ”.
Thus, in the field of natural language processing (NLP), the submitted manuscript is about extracting Named Entity recognition (NER), more precisely about Nested Named Entity Recognition (NNER), such as names, locations, companies, events, products, themes, topics. This recognition-identification processing necessitates to locate and classify hierarchically structured semantic named entities mentioned in texts.
For learning about NNER semantic information from texts, the manuscript is in the line of Strubell et al. proposing an Induced Dilated Convolutional Neural Network (IDCNN) instead of Convolutional Neural Networks (CNNs) and of Katiyar et al. using “a recursive neural network to extract features from directed hypergraph representations and constructed an LSTM-based sequence labeling model to learn the hypergraph representation of nested entities in the text”.
Their Induced Dilated Convolutional Neural Network LeakyReLU (IDCNNLR) semantic extraction module was used with two datasets that are CMeEE (Chinese medical texts) and CLUENER2020 (an efficient chinese text classifier, financial corpus) datasets with 9 and 10 label types, respectively.
Evaluation of models is correctly done using the classical signal detection theory (SDT) with proportion of Hits, False Alarms, Misses and Correct Rejections and derive the four classical measures of detection performance of systems [Accuracy] (Precision, Recall, specificity and Sensitivity). This research uses precision and recall to get the F-Measure to evaluate the performance of the model. These results were compared to the results of other models in the literature. In addition, the contribution of each of the modules
To be corrected
- The authors should inform readers about the meaning of acronyms at the first use: please introduce the meanings of acronyms of words before using them.
- References within the list are of different formats; please correct
- Provide the link for access to the annotated datasets (both CMeEE and CLUENER2020)
- « CLUENER2020 [32] is a Chinese fine-grained named entity » is a copy/past from “Li, W., Liu, J., Gao, Y., Zhang, X., & Gu, J. (2023). Chinese Fine-Grained Named Entity Recognition Based on BILTAR and GlobalPointer Modules. Applied Sciences, 13(23), 12845 ». Please correct by CLUENER2020 [25].
- Lines 477 to 505 are also copy-past from Li, W., Liu, J., Gao, Y., Zhang, X., & Gu, J. (2023). Please rewrite.
- Please complete translation in Chinese exemples :” and "北京" is a location 434 entity. Both entities are overlapped in the text segment "北京大å¦."

Author Response
The modification content is attached.

Reviewer 3 Report
Comments and Suggestions for Authors
The authors propose a new model called BERT-ATT-IDCNNLR-ATTR-GlobalPointer to identify nested entities in text. The model is based on an IDCNNLR module which extracts semantic information and uses multiple-head self-attention mechanisms at multiple positions to achieve data denoising. Based on that information, a GlobalPointer module considers each possible entity boundary. The BERT model is used for word embedding. The effectiveness of the proposed model was evaluated on two Chinese entity recognition datasets. The model achieved F1 scores of 69.617% and 79.285% on the CMeEE and CLUENER2020 datasets, respectively. A comparison with other models from the state-of-the-art showed the performance improvement of the proposed model.
The approach taken by the authors is sound. The paper is well written and clear.
However, the following points must be addressed:
1. Equations and symbols must be written using a math font instead of images.
2. Typos and spelling errors annotated in the PDF file must be corrected.
3. Some details about the proposed model are missing. Please include the total number of parameters, the size of the input, the size of each layer, the size of each kernel, etc.
4. Some details about the training conditions of the model are missing. For instance, include the loss function.
5. Some details about the implementation are missing. Please include details such as type of computer, operating system, processor, details about the GPU used, etc.
6. Details about the datasets are missing. Include the number of samples per class, type of cross-validation employed, the size of the validation/testing dataset, etc.

Comments on the Quality of English Language
In general English language is good. However some minor corrections are necessary. Please find some annotations in the included PDF file asi-2754183-peer-reviewed-v1.pdf.
Author Response
The modification content is attached.
